

# The core clock gene *Per1* phases molecular and electrical circadian rhythms in SCN neurons

Jeff R. Jones[1,3] and Douglas G. McMahon[1,2]

[1] Neuroscience Graduate Program, Vanderbilt University, Nashville, TN, United States
[2] Department of Biological Sciences, Vanderbilt University, Nashville, TN, United States
[3] Current affiliation: Department of Biology, Washington University in St. Louis, St. Louis, MO, United States

## ABSTRACT

The brain's biological clock, the suprachiasmatic nucleus (SCN), exhibits endogenous 24-hour rhythms in gene expression and spontaneous firing rate; however, the functional relationship between these neuronal rhythms is not fully understood. Here, we used a *Per1*::GFP transgenic mouse line that allows for the simultaneous quantification of molecular clock state and firing rate in SCN neurons to examine the relationship between these key components of the circadian clock. We find that there is a stable, phased relationship between E-box-driven clock gene expression and spontaneous firing rate in SCN neurons and that these relationships are independent of light input onto the system or of GABA$_A$ receptor-mediated synaptic activity. Importantly, the concordant phasing of gene and neural rhythms is disrupted in the absence of the homologous clock gene *Per1*, but persists in the absence of the core clock gene *Per2*. These results suggest that *Per1* plays a unique, non-redundant role in phasing gene expression and firing rate rhythms in SCN neurons to increase the robustness of cellular timekeeping.

## INTRODUCTION

Understanding how gene signaling networks influence the activity of neurons and circuits that control behavior is an essential problem in neuroscience. Daily changes in physiology and behavior in mammals are driven by the suprachiasmatic nucleus (SCN), a network of neurons exhibiting endogenous rhythms in gene expression and firing rate in isolation (*Colwell, 2011*). A key unsolved question in circadian neurobiology is how these neuronal rhythms interact to form a coherent pacemaker. Circadian gene expression rhythms in SCN neurons are driven by an autoregulatory transcriptional/translational feedback loop (TTFL) comprised of the core clock genes *Bmal1*, *Clock*, *Per1/2*, and *Cry1/2*. Furthermore, circadian firing rate rhythms are produced by a collection of intrinsic ionic currents that allow SCN neurons to spontaneously fire action potentials and, importantly, modulate their firing rates so that they fire at up to 6–12 Hz during the day and 0–2 Hz at night. The central role of the TTFL in driving circadian electrical activity has been supported by studies in which the cycling of the molecular clock was halted by the elimination of critical clock genes (*Liu et al., 1997*; *Herzog, Takahashi & Block, 1998*; *Nakamura et al., 2002*; *Albus et al., 2002*). However, blocking action potentials in SCN neurons and in clock neurons during

Corresponding author
Douglas G. McMahon,
douglas.g.mcmahon@vanderbilt.edu

development in *Drosophila* disrupts gene cycling (*Yamaguchi et al., 2003*; *Nitabach, Blau & Holmes, 2002*), and optogenetically driving SCN neuron firing rates can reset the molecular clockworks (*Jones, Tackenberg & McMahon, 2015*). Thus, the relationship between clock neuron electrical activity and the molecular clockworks is likely bidirectional.

A fundamental gap in our knowledge is therefore understanding how the circadian molecular clockworks is linked to firing rate rhythms in the SCN. To investigate this, we have used our *Per1*::GFP transgenic mouse line in which sequences from the *Per1* gene promoter that contain both E-box enhancer elements for CLOCK/BMAL1-driven transcription as well as CRE elements for CREB-driven induction drive a short half-life version of enhanced green fluorescent protein (d2EGFP; *Kuhlman, Quintero & McMahon, 2000*). Importantly, there is both a high concordance of d2EGFP expression with *Per1* mRNA and PER1 protein expression in the SCN of these mice and congruous regional distributions and rhythms which suggests that the d2EGFP construct in *Per1*::GFP mice reports native *Per1* gene expression with high fidelity (*LeSauter et al., 2003*). Using this artificial clock-controlled gene, we have previously shown that the degree of activation of this construct correlates with firing rate in individual SCN neurons during the day phase of circadian cycling and following a phase-resetting light pulse at night (*Kuhlman et al., 2003*; *Quintero, Kuhlman & McMahon, 2003*). This correlation and additional experiments with circadian reporters suggest that a fixed phase relationship may exist between the molecular clockworks and circadian electrical activity (*Colwell, 2011*). Intriguingly, neuropeptide resetting of SCN neuron firing rate requires the translation of the native *Per1* gene, which suggests a functional role for *Per1* in this relationship (*Gamble et al., 2007*; *Kudo et al., 2013*).

In order to study *Per1*'s role as a potential link between molecular and electrical circadian rhythms, we crossed our *Per1*::GFP mice with *Per1*$^{-/-}$, *Per2*$^{-/-}$, and *Per1*$^{-/-}$; *Per2*$^{-/-}$ mice (*Bae et al., 2001*) that had been bred congenic on a C57BL/6J background (*Pendergast, Friday & Yamazaki, 2009*; *Pendergast, Friday & Yamazaki, 2010a*; *Pendergast, Friday & Yamazaki, 2010b*) to yield *Per* knockout mice that express d2EGFP as a transcriptional reporter of the molecular clockworks. Importantly, the GFP construct is still rhythmically expressed in *Per1*$^{-/-}$ *Per1*::GFP animals because its production is regulated by the E-box and CRE elements in the *Per1* promoter sequences in the transgene (*Kuhlman, Quintero & McMahon, 2000*). *In vivo*, the knockout of *Per1* or *Per2* individually in mice on a congenic C57BL/6J background results in rhythmic wheel-running behavior that is similar to that of wild-type mice (*Pendergast, Friday & Yamazaki, 2009*; *Pendergast, Friday & Yamazaki, 2010a*; *Xu et al., 2007*), while *Per1*$^{-/-}$; *Per2*$^{-/-}$ double knockout mice on a C57BL/6J background are behaviorally arrhythmic in constant darkness (*Pendergast & Yamazaki, 2011*). *In vitro*, however, SCN cultures from *Per1*$^{-/-}$ mice exhibit weakened or variable rhythms in circadian gene expression as read out by PER2::LUCIFERASE that can, in some cases, be enhanced by a media change (*Liu et al., 2007*; *Pendergast, Friday & Yamazaki, 2010a*; *Pendergast, Friday & Yamazaki, 2010b*; *Ruan et al., 2012*). The period of *Per1-luc* gene expression rhythms in *Per2*$^{-/-}$ SCN cultures is markedly shorter than the period of wild-type SCN rhythms (*Pendergast, Friday & Yamazaki, 2010a*). Importantly, *in vivo* multi-unit neural activity from *Per1*$^{-/-}$ mice is rhythmic, which may explain the discrepency between the

weakly-rhythmic molecular clockworks observed *in vitro* and robustly rhythmic behavioral rhythms observed *in vivo* in these mice (*Takasu et al., 2013*).

Here we report a consistent, almost synchronous, phase relationship between the molecular clockworks and circadian electrical activity throughout the 24 h circadian day assayed in individual SCN neurons of *in vitro* SCN slices. Knockout of *Per1* alters this relationship, delaying molecular activity relative to firing rate rhythms by about 6 h such that there is an approximately 90 degree phase angle between them. Importantly, this effect is specific to *Per1*, as the close synchrony between circadian gene expression and firing rate rhythms is preserved when the clock gene *Per2* is knocked out. Taken together, these results identify *Per1* as a clock gene that phases the molecular clockworks and circadian electrical activity into reinforcing synchrony.

## MATERIALS & METHODS

### Animals and housing
Experiments were performed using male and female *Per1*::d2EGFP ("*Per1*::GFP"; *Kuhlman, Quintero & McMahon, 2000*), *Per1*::GFP x *Per1*$^{-/-}$ (*Bae et al., 2001*; "*Per1*$^{-/-}$"), *Per1*::GFP x *Per2*$^{-/-}$ (*Bae et al., 2001*; "*Per2*$^{-/-}$"), or *Per1*::GFP x *Per1*$^{-/-}$ x *Per2*$^{-/-}$ ("*Per1*$^{-/-}$; *Per2*$^{-/-}$") mice on a C57BL/6J background, 1–3 months of age. Animals were provided with food and water *ad libitum* and were housed in single-sexed cages of no more than five animals from weaning until experimental use on a 12:12 light/dark (LD) cycle, or, for some experiments, were housed in constant darkness (DD) for at least two weeks before use. All animal care and experimental procedures were conducted in concordance with Vanderbilt University's Institutional Animal Care and Use Committee guidelines.

### Behavioral analysis
Wheel-running activity from mice housed in DD was monitored and recorded in 5 min bins using ClockLab software (Actimetrics, Evanston, IL, USA). Time of activity onset (defined as circadian time 12 [CT 12]) was determined using ClockLab Analysis Software. As *Per1*$^{-/-}$; *Per2*$^{-/-}$ mice are behaviorally arrhythmic, CT could not be defined for this group of mice.

### Slice preparation
Mice housed in LD were killed by cervical dislocation without anesthesia in ambient light for dissections occurring between ZT (Zeitgeber time) 0–12, where ZT 0 is defined as the time of lights on and ZT 12 is defined as the time of lights off. For dissections occurring between ZT 12–24 for mice housed in LD, or for all dissections from mice housed in DD, mice were killed by cervical dislocation without anesthesia under dim red light (<1 lux). Importantly, dissections occurring in the dark phase have been shown not to affect the phase of electrical activity rhythms in the SCN (*VanderLeest et al., 2009*). After dissection, brains were quickly removed and blocked in cold, oxygenated 95% $O_2$–5% $CO_2$ dissecting solution (in mM: 114.5 NaCl, 3.5 KCl, 1 $NaH_2PO_4$, 1.3 $MgSO_4$, 2.5 $CaCl_2$, 10 D-glucose, and 35.7 $NaCHO_3$). SCN slices (200 $\mu$m) were cut coronally on a vibroslicer (World Precision Instruments, Sarasota, FL, USA) at 4–10 °C and transferred directly to an
open recording chamber continually superfused with warmed (35 ± 0.5 °C) extracellular solution (in mM: 124 NaCl, 3.5 KCl, 1 NaH$_2$PO$_4$, 1.3 MgSO$_4$, 2.5 CaCl$_2$, 10 D-glucose, and 26 NaCHO$_3$). Slices were allowed to recover for 1 h before recording.

### Electrophysiological recording and imaging

SCN neurons were visualized using a Leica DMLFS microscope (Leica Microsystems, Buffalo Grove, IL, USA) equipped with near-infrared (IR)-differential interference contrast and fluorescence optics. For loose-patch recordings, patch electrodes (4–6 MΩ) pulled from glass capillaries (WPI) on a multistage puller (DMZ; Zeitz, Martinsried, Germany) were filled with extracellular solution. Spontaneous action potential recordings (∼5 min in duration) from neurons sampled throughout the SCN were obtained with an Axopatch 200 B amplifier (Molecular Devices, Sunnyvale, CA, USA) and monitored online with pClamp 10.0 software (Molecular Devices). Recordings obtained in gap-free mode throughout the day were sampled at 10 kHz, and were filtered online at 1 kHz. Loose-patch seal resistances ranged from 10–30 MΩ. Slices were used for no more than 6 h after dissection. Immediately after cessation of electrophysiological recording, an image of the recorded neuron was captured with an exposure time of one second using HCImage acquisition software (Hamamatsu Photonics, Bridgewater, NJ, USA) with a cooled CCD camera (Hamamatsu) and an EGFP filter set. All recordings were confirmed to be from GFP$^+$ neurons by aligning digital images of the same neuron taken under near-IR and GFP fluorescence illumination.

### Image and electrophysiology analysis

Image analysis was performed based on methods described in *Kuhlman et al. (2003)*, using ImageJ software with 16-bit digitization. Fluorescence was reported as the intensity of a region of interest containing the cell body divided by the background fluorescence to normalize for differences in baseline fluorescence across preparations and fields. Background fluorescence was defined as the average pixel intensity of two local measurements next to the recorded neuron and the total frame (1,024 × 1,024 pixels). Average firing rate for the recording period was calculated using Clampex software (Molecular Devices).

## RESULTS

To assay the phase relationship between E-box-driven clock gene expression and spontaneous firing rate in the SCN, we performed visually-targeted loose-patch recording of individual GFP$^+$ SCN neurons in coronal SCN slices from *Per1*::GFP mice (*Kuhlman, Quintero & McMahon, 2000*) sampling around the clock. We found that both E-box-driven clock gene expression as measured by GFP fluorescence intensity and spontaneous firing rate showed temporal variations consistent with ongoing rhythms at a population level (Fig. 1A). The population of recorded SCN neurons exhibited a peak in spontaneous firing rate as determined by a Cosinor fit at around Zeitgeber Time (ZT) 7 and a peak in GFP fluorescence intensity at ZT 9, where ZT 0 is defined as the time of lights on. Thus, observed rhythms in firing rate and gene expression were closely aligned, with firing rate phase leading by about two hours. When the time to translate and fold d2EGFP of

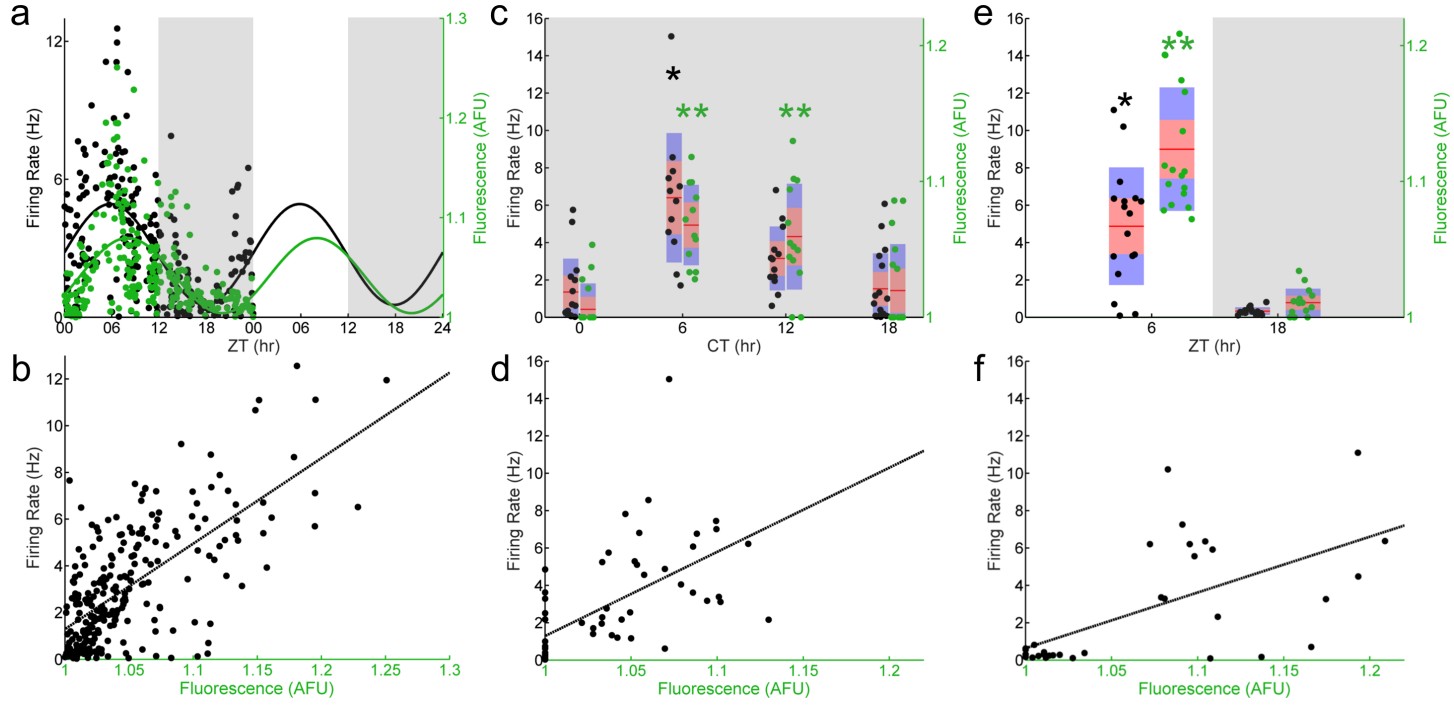

**Figure 1  E-box driven gene expression and spontaneous firing rate are rhythmic and correlated in individual SCN neurons.** (A) LD (light, white shading; dark, gray shading) firing rates (black dots) and fluorescence intensities (green dots) from individual *Per1*::GFP SCN neurons recorded throughout the day ($n = 249$ cells, 34 mice). Population rhythmicity determined using a Cosinor fit; firing rate (black line), $r^2 = 0.3267$, $p < 0.001$; fluorescence intensity (green line), $r^2 = 0.2985$, $p < 0.001$. (B) Firing rate versus fluorescence intensity in individual *Per1*::GFP SCN neurons recorded throughout the day in LD show a positive, linear correlation ($n = 249$ cells, 34 mice; Pearson's $r$, $r^2 = 0.4504$, $p < 0.001$). (C) DD (gray shading) firing rates (black dots) and fluorescence intensities (green dots) from individual *Per1*::GFP SCN neurons recorded throughout the day ($n = 58$ cells, 10 mice). Population rhythmicity determined using a Kruskal–Wallis ANOVA on Ranks test (∗, firing rate, $p < 0.001$; ∗∗, fluorescence, $p < 0.001$). Points are overlaid with red (95% confidence interval) and blue bars (one standard deviation). (D) Firing rate versus fluorescence intensity in individual *Per1*::GFP SCN neurons recorded throughout the day in DD show a positive, linear correlation ($n = 58$ cells, 10 mice; Pearson's $r$, $r^2 = 0.3621$, $p < 0.001$). (E) LD firing rates (black dots) and fluorescence intensities (green dots) from individual *Per1*::GFP SCN neurons recorded throughout the day in the presence of GABAzine ($n = 32$ cells, 4 mice). Population rhythmicity determined using a Mann–Whitney $U$ test (∗, firing rate, $p < 0.001$; ∗∗, fluorescence, $p < 0.001$). Points are overlaid with red (95% confidence interval) and blue bars (one standard deviation). (F) Firing rate versus fluorescence intensity in individual *Per1*::GFP SCN neurons recorded throughout the day in LD in the presence of GABAzine show a positive, linear correlation ($n = 32$ cells, 4 mice; Pearson's $r$, $r^2 = 0.5744$, $p < 0.001$).

approximately two hours is accounted for (*Li et al., 1998*), the firing rate rhythm would be projected to be essentially synchronous with the transcriptional rhythm. To further analyze the relationship between GFP fluorescence intensity and spontaneous firing rate we plotted these parameters for each recorded neuron (Fig. 1B). Overall, fluorescence intensity ranged from 1.0 to 1.25x background, while firing rates varied from 0 to 12 Hz. Individual SCN neurons showed a range of variability in both firing rates and fluorescence intensities, and in aggregate there was a significant positive linear correlation between *Per1* promoter activiation as read out by GFP fluorescence intensity and spontaneous firing rate assayed within individual SCN neurons over the course of the 24 h sampling (Fig. 1B; $n = 249$ cells, 34 mice; Pearson's $r$, $r^2 = 0.4504$, $p < 0.001$). Although this relationship can be approximated as linear, it is likely to be more complex and cyclical in nature (see 'Discussion').

To test whether these results were due to circadian rhythms or diurnal light-driven responses we performed experiments on SCN from mice housed in constant darkness. Both GFP fluorescence intensity and spontaneous firing rate remained rhythmic (Fig. 1C), and the positive correlation between GFP fluorescence intensity and spontaneous firing rate in individual neurons persisted in constant darkness (Fig. 1D). Finally, most SCN neurons are GABAergic (*Wagner et al., 1997*) and therefore GABA transmission represents much of the ongoing rapid synaptic transmission in the nucleus. We found that when the GABA$_A$-receptor blocker GABAzine was applied to SCN neurons from mice housed under a normal light cycle, GFP fluorescence intensity and spontaneous firing rate remained rhythmic at the population level (Fig. 1E). Likewise, the positive correlation between GFP fluorescence intensity and spontaneous firing rate in individual neurons again persisted in the presence of this blocker (Fig. 1F).

To determine if this observed population- and single-cell stable phase relationship between E-box-driven gene expression and spontaneous firing rate persisted in the absence of the core clock gene *Per1*, we recorded individual GFP$^+$ SCN neurons from *Per1*$^{-/-}$; *Per1*::GFP mice sampled around the clock. Importantly, in these mice, the E-box-driven production of GFP still acts as a reporter of the molecular clockworks since the reporter gene is driven by a BMAL1/CLOCK heterodimer even in the absence of the native *Per1* gene. We found that at the population level both E-box-driven clock gene expression as measured by GFP fluorescence intensity and spontaneous firing rate exhibited statistically significant time series variations, however, their phase relationship was radically changed (Fig. 2A). SCN neurons from *Per1*$^{-/-}$ mice exhibited a peak in GFP fluorescence intensity as determined by a Cosinor fit that was significantly delayed (ZT 12) compared to the peak of spontaneous firing rate (ZT 5)—a difference of about 7 h. Strikingly, the robust correlation between GFP fluorescence intensity and spontaneous firing rate we observed in individual SCN neurons from wild-type mice was completely lost in individual *Per1*$^{-/-}$ SCN neurons (Fig. 2B). This is an outcome of the change in relative phases of the firing rate and molecular rhythms to an approximately 6 h (i.e., 90 degree) difference. In SCN neurons from *Per1*$^{-/-}$ mice housed in constant darkness, the correlation between GFP fluorescence intensity and spontaneous firing rate in individual cells remained absent (Fig. 2D); at the population level, GFP fluorescence intensity remained rhythmic, and the temporal profile of the spontaneous firing rate was below the significance level for rhythmicity ($p = 0.067$; Fig. 2C).

To determine if the altered population-level and abolished single-cell phase relationship between E-box-driven gene expression and spontaneous firing rate we observed in *Per1*$^{-/-}$ SCN neurons was specific to the loss of *Per1*, we recorded from individual GFP$^+$ SCN neurons in SCN slices from *Per2*$^{-/-}$; *Per1*::GFP mice sampled throughout the day. Both E-box-driven clock gene expression as measured by GFP fluorescence intensity and spontaneous firing rate were rhythmic at a population level (Fig. 3A). Similar to wild type, and in contrast to *Per1*$^{-/-}$ SCN neurons, the population of recorded SCN neurons in *Per2*$^{-/-}$ mice exhibited a peak in spontaneous firing rate at about ZT 7 and a peak in GFP fluorescence intensity at about ZT 9. Critically, the single-cell correlation between GFP fluorescence intensity and spontaneous firing rate was also preserved in the absence

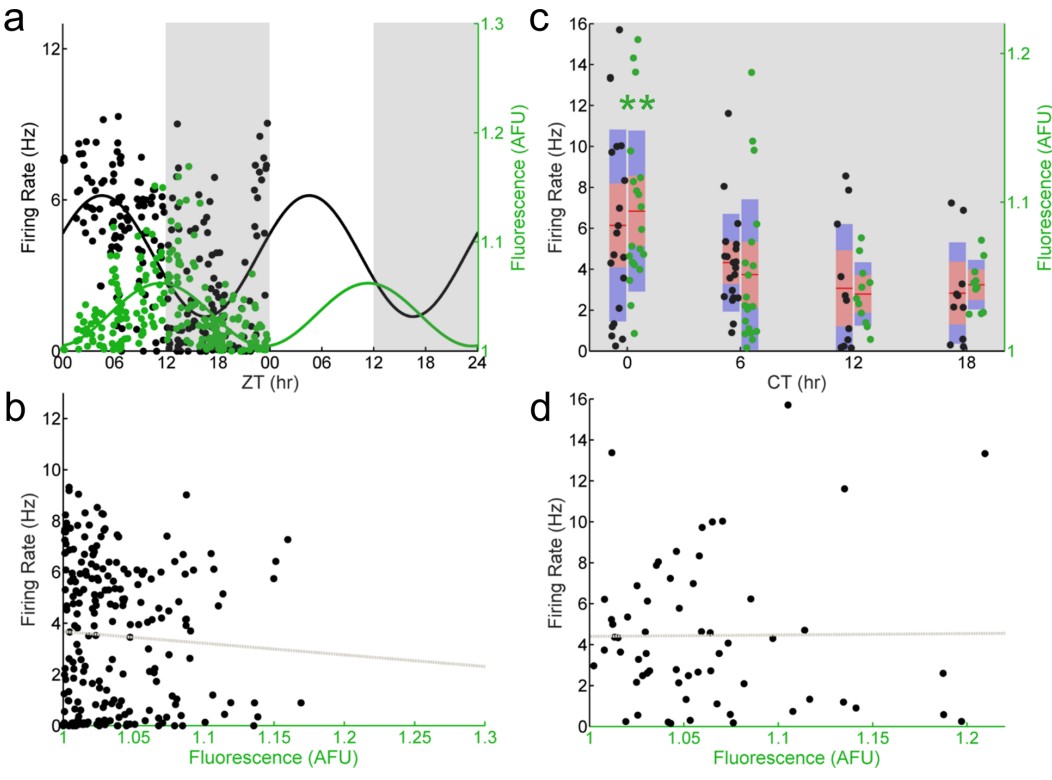

**Figure 2** **E-box driven gene expression and spontaneous firing rate are not correlated in individual** *Per1–/–* **SCN neurons.** (A) LD (light, white shading; dark, gray shading) firing rates (black dots) and fluorescence intensities (green dots) from individual $Per1^{-/-}$; $Per1$::GFP SCN neurons recorded throughout the day ($n = 227$ cells, 21 mice). Population rhythmicity determined using a Cosinor fit; firing rate (black line), $r^2 = 0.3988$, $p < 0.001$; fluorescence intensity (green line), $r^2 = 0.2822$, $p < 0.001$. (B) Firing rate versus fluorescence intensity in individual $Per1^{-/-}$ SCN neurons recorded throughout the day in LD are not correlated ($n = 227$ cells, 21 mice; Pearson's $r$, $r^2 = 0.0032$, $p = 0.3968$). (C) DD (gray shading) firing rates (black dots) and fluorescence intensities (green dots) from individual $Per1^{-/-}$ SCN neurons recorded throughout the day ($n = 61$ cells, 7 mice). Population rhythmicity determined using a Kruskal–Wallis ANOVA on Ranks test (∗, firing rate, $p = 0.067$; ∗∗, fluorescence, $p < 0.001$). Points are overlaid with red (95% confidence interval) and blue bars (one standard deviation). (D) Firing rate versus fluorescence intensity in individual $Per1^{-/-}$ SCN neurons recorded throughout the day in DD are not correlated ($n = 61$ cells, 7 mice; Pearson's $r$, $r^2 = 0.0049$, $p = 0.4831$).

of *Per2* (Fig. 3B). In SCN neurons from $Per2^{-/-}$ mice housed in constant darkness, we continued to observe stable population-level and single-cell phase relationships between GFP fluorescence intensity and spontaneous firing rate (Figs. 3C and 3D). This single-cell correlation was abolished when *Per1* was concurrently knocked out in SCN neurons from $Per1^{-/-}$; $Per2^{-/-}$ mice (Fig. 4).

## DISCUSSION

To investigate the relationship between gene expression rhythms and circadian electrical activity in the SCN, we performed firing rate recording and real-time fluorescence imaging of the state of the molecular clockworks in $Per1$::GFP$^+$ SCN neurons throughout the circadian day. We found that E-box-driven gene expression and spontaneous firing rate rhythms were closely aligned in individual $Per1$::GFP SCN neurons, with firing

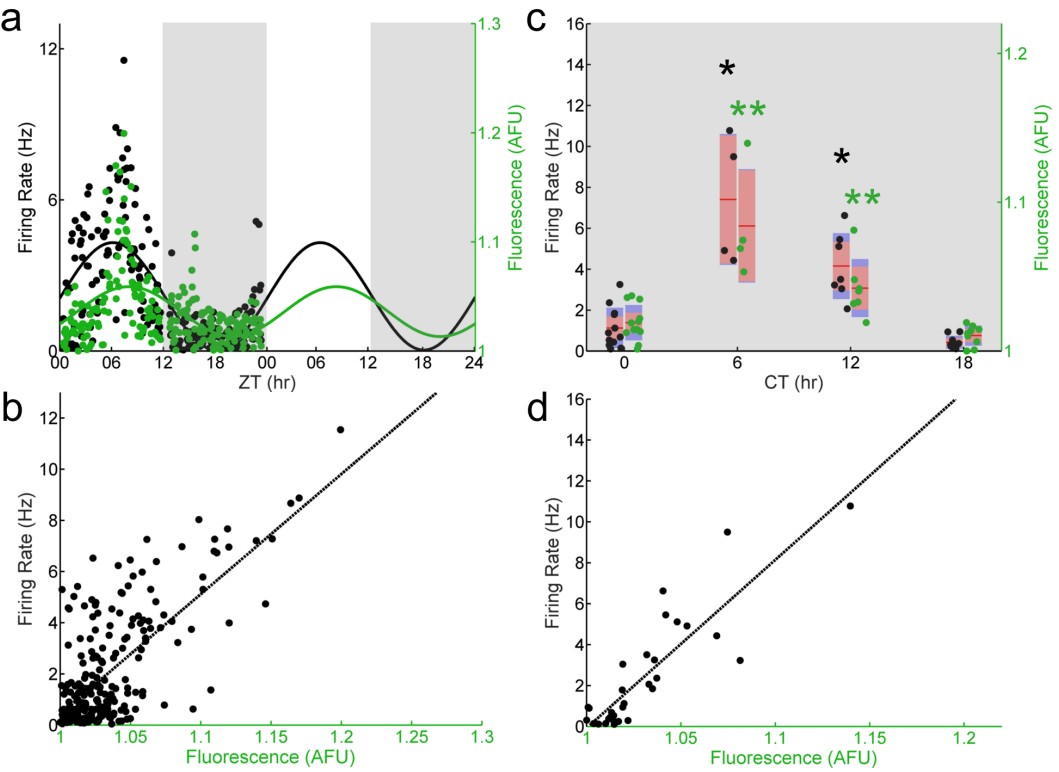

**Figure 3** **The correlation between E-box driven gene expression and spontaneous firing rate is preserved in individual *Per2–/–* SCN neurons.** (A) LD (light, white shading; dark, gray shading) firing rates (black dots) and fluorescence intensities (green dots) from individual *Per2*$^{–/–}$; *Per1*::GFP SCN neurons recorded throughout the day ($n = 230$ cells, 10 mice). Population rhythmicity determined using a Cosinor fit; firing rate (black line), $r^2 = 0.4951$, $p < 0.001$; fluorescence intensity (green line), $r^2 = 0.2561$, $p < 0.001$. (B) Firing rate versus fluorescence intensity in individual *Per2*$^{–/–}$ SCN neurons recorded throughout the day in LD show a positive, linear correlation ($n = 230$ cells, 10 mice; Pearson's $r$, $r^2 = 0.6328$, $p < 0.001$). (C) DD (gray shading) firing rates (black dots) and fluorescence intensities (green dots) from individual *Per2*$^{–/–}$ SCN neurons recorded throughout the day ($n = 32$ cells, 7 mice). Population rhythmicity determined using a Kruskal-Wallis ANOVA on Ranks test (∗, firing rate, $p < 0.001$; ∗∗, fluorescence, $p < 0.001$). Points are overlaid with red (95% confidence interval) and blue bars (one standard deviation). (D) Firing rate versus fluorescence intensity in individual *Per2*$^{–/–}$ SCN neurons recorded throughout the day in DD show a positive, linear correlation ($n = 32$ cells, 7 mice; Pearson's $r$, $r^2 = 0.7572$, $p < 0.001$).

rate rhythms consistently phase leading gene expression rhythms by about two hours. Given the estimated two hour time course for the translation and folding of d2EGFP (*Li et al., 1998*), this suggests that spontaneous firing and the activation of the *Per1* promoter are essentially synchronous. In SCN neurons from *Per1*$^{–/–}$ mice this relationship is altered such that gene expression rhythms are significantly delayed in relation to firing rate rhythms. Importantly, in the absence of *Per1* there is no longer a predictive correlation between gene expression and neural activity in individual SCN neurons. However, in SCN neurons from *Per2*$^{–/–}$ mice the phase relationship between circadian rhythms in gene expression and firing rate is intact. These results therefore demonstrate that *Per1* is necessary for maintaining a synchronous phase relationship between the molecular clockworks and membrane electrical activity.

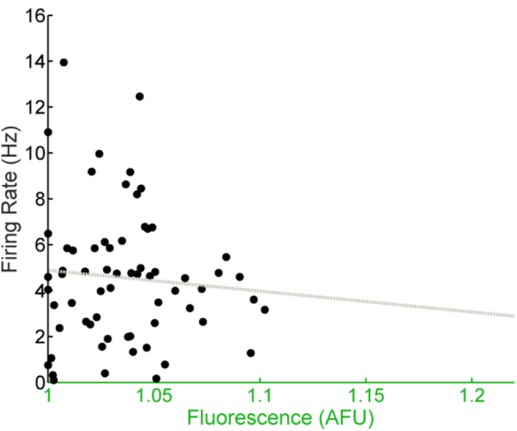

**Figure 4** **E-box driven gene expression and spontaneous firing rate are not correlated in individual *Per1–/–; Per2–/–* SCN neurons.** Firing rate versus fluorescence intensity in individual $Per1^{-/-}$; $Per2^{-/-}$ SCN neurons recorded throughout the day in DD are not correlated ($n = 65$ cells, 6 mice; Pearson's $r$, $r^2 = 0.0036$, $p = 0.5146$).

The finding that there exists a stable, reinforcing phase relationship between E-box driven gene expression and spontaneous firing rate across an entire circadian day expands upon previous studies showing that there is a correlation between these two components of the circadian clock at midday and after a phase-resetting light pulse at night (*Kuhlman et al., 2003*; *Quintero, Kuhlman & McMahon, 2003*). Here we not only show that this correlation persists in neurons sampled across 24 h, but that gene expression rhythms consistently aligned in phase with spontaneous firing rate rhythms. Importantly, *Per1*::GFP expression in a wild-type animal peaks at approximately the same time as the peak of native PER1 protein expression (*LeSauter et al., 2003*; which peaks approximately two to four hours after the peak of native *Per1* mRNA expression), and our observed time of peak spontaneous firing rate is also consistent with that found in other studies (*Meijer et al., 1997*; but see *Belle et al., 2009*). Thus, by simultaneously measuring the state of the molecular clock and electrical activity in individual neurons within SCN slices, we were able to determine the instantaneous circadian relationship between these two key components of the circadian clock in individual SCN neurons. This newly established within-cell relationship therefore predicts that an increase in firing rate precedes an increase in translation of PER1 (although it may occur virtually concurrently with the transcription of *Per1* itself). The canonical multi-component model of the mammalian circadian clock supposes that firing rate is solely an output of the state of the molecular clock; however, these results suggest that firing rate, peaking ∼2 h before the peak of GFP fluorescence (and thus the peak of PER1 translation), is likely an input onto the molecular clock, as also suggested by recent results demonstrating that optogenetic manipulation of clock neurons can reset the molecular clockworks (*Jones, Tackenberg & McMahon, 2015*).

Our $Per1^{-/-}$ data suggest that the alignment of SCN neuronal firing rate and the molecular clockworks in a reinforcing phase relationship increases the robustness of both oscillations. Knocking out *Per1* results in a 90 degree phase relationship between these components of the circadian clock that weakens molecular rhythms as evidenced by a decrease in the

amplitude of *Per1*::GFP fluorescence rhythms in LD and a lack of *Per1*::GFP fluorescence rhythms in DD. Interestingly, the weakened neural and molecular rhythms we observe in *Per1*$^{-/-}$ SCN neurons may account for the high-amplitude phase response curve to light observed *in vivo* in *Per1*$^{-/-}$ mice, as a weaker central clock is shifted more easily (*Pendergast, Friday & Yamazaki, 2010b*).

The *Per1*$^{-/-}$ and *Per2*$^{-/-}$ data presented here are in agreement with previously reported data from these mice on a congenic C57BL/6J background. *Per1*$^{-/-}$ mice are behaviorally rhythmic (*Pendergast, Friday & Yamazaki, 2009*), exhibit *in vivo* SCN firing rate rhythms (*Takasu et al., 2013*) and, importantly, their *in vitro* SCN molecular rhythms as assayed by *Per1-luc* are less robustly rhythmic for one or more peaks *in vitro* with a delayed phase (*Pendergast, Friday & Yamazaki, 2009*); similarly, *Per2*$^{-/-}$ mice have intact behavioral and molecular rhythms (*Pendergast, Friday & Yamazaki, 2010a*; *Pendergast, Friday & Yamazaki, 2010b*). In the present study, SCN neurons in SCN slices from our *Per1*$^{-/-}$ mice as a population exhibit somewhat depressed firing rate rhythms and rhythmic but dampened and phase-delayed *Per1*::GFP fluorescence rhythms, while SCN neurons from our *Per2*$^{-/-}$ mice exhibit firing rate and fluorescence rhythms that are comparable to those of wild-type controls.

How, though, could clock neuron electrical activity, an ostensible output of the molecular clock, also affect the state of the molecular clock itself? The most likely candidate for this connection is yet another component of neuronal circadian rhythms—daily oscillations in intracellular second messengers, including $Ca^{2+}$ and cAMP (*Brancaccio et al., 2013*). Some SCN neurons can exhibit firing rate rhythms on genetic backgrounds in which gene cycling is absent, suggesting that the ionic mechanisms at the membrane can oscillate in a circadian manner without an intact TTFL (*Nakamura et al., 2002*). Similarly, genetic or physiological blockade of the firing rate rhythm in *Drosophila* (dORK channel) or mouse (TTX) leads to run-down of clock gene cycling in clock neurons (*Nitabach, Blau & Holmes, 2002*; *Yamaguchi et al., 2003*). This is likely due to blunting of cellular calcium rhythms as chelation of extracellular $Ca^{2+}$ blunts gene expression rhythms (*Lundkvist et al., 2005*). Interestingly, a recent study using antisense oligodeoxynucleotides to knock down *Per1* in SCN neurons found that this treatment not only transiently alters firing rate but also reduces intracellular levels of calcium and levels of calcium-activated potassium currents (*Kudo, Block & Colwell, 2015*). Finally, another intracellular messenger, cAMP, which in SCN neurons is primarily controlled by VIPergic intercellular communication through VPAC2 receptors, is also critical for sustained gene and firing rate rhythms (*Aton et al., 2005*; *Atkinson et al., 2011*; *Cutler et al., 2003*).

We suggest that *Per1*, with its rapid high-amplitude CRE-driven responses (*Tischkau et al., 2003*), mediates resonant phasing of firing rate and E-box-driven transcription in both wild-type and *Per2*$^{-/-}$ SCN neurons. This phasing likely occurs through the rapid response of *Per1* to CREB/CRE signals from upstream membrane-driven $Ca^{2+}$ and/or cAMP (*Brancaccio et al., 2013*) and subsequent downstream inhibition of potassium currents through FDR or other potassium channels (*Kuhlman & McMahon, 2004*; *Kudo et al., 2011*). Without *Per1*, the rhythms in firing rate and gene expression are displaced out of near-synchronous phase to a 90 degree relative phase where they are

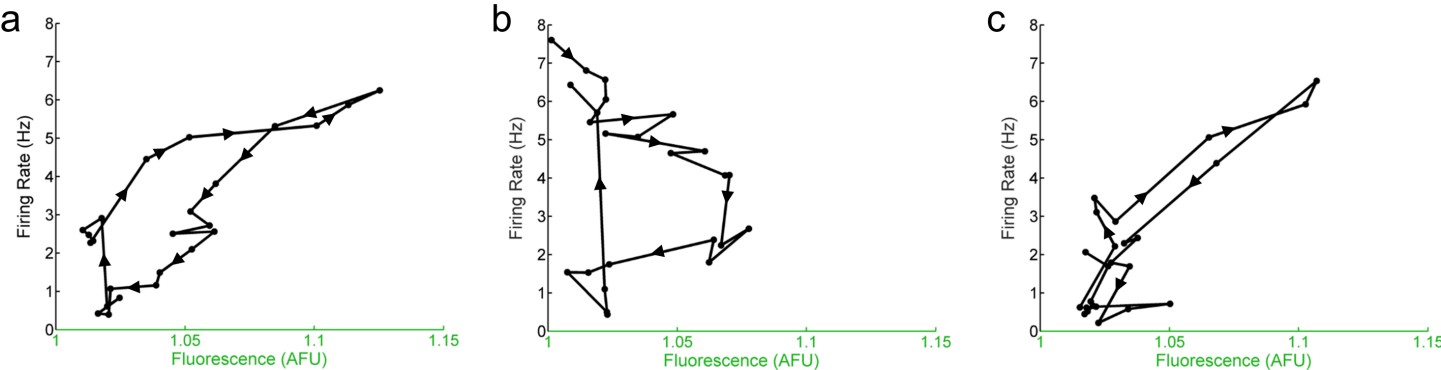

**Figure 5** **The hourly relationship between E-box driven gene expression and spontaneous firing rate is cyclical.** Mean hourly firing rate and fluorescence intensities (black dots) of *Per1*::GFP SCN neurons recorded throughout the day in LD. (B) Mean hourly firing rate and fluorescence intensities (black dots) of *Per1*[−/−] SCN neurons recorded throughout the day in LD. (C) Mean hourly firing rate and fluorescence intensities (black dots) of *Per2*[−/−] SCN neurons recorded throughout the day in LD. In (A–C), arrows and lines represent the direction of the progression of the time of day of recording (i.e., from ZT 0 to ZT 23).

no longer reinforcing. As a consequence, single-cell SCN oscillations in *Per1*[−/−] mice are weakened (*Pendergast, Friday & Yamazaki, 2009*). *In situ*, in the fully intact SCN network, this effect can be rescued through SCN network coupling; however, in dispersed SCN neurons, in fibroblasts, and in peripheral tissue cells that lack the SCN's inherent network coupling, knocking out *Per1* strongly disrupts the function of the molecular clock (*Liu et al., 2007*). A limitation of our current study is that it was performed in SCN slices, which represent a semi-intact network. Thus, we cannot on the basis of this data determine which aspects of the altered SCN neuron function in *Per1*[−/−] mice are due to cell-autonomous effects and which may be due to network compensation. Further experiments with dispersed SCN neurons are necessary to resolve this issue.

Intriguingly, when the relationship between firing rate and the molecular clock in individual SCN neurons from *Per1*::GFP, *Per1*[−/−], and *Per2*–/– mice is analyzed over time, a cyclical pattern emerges (Fig. 5). Although there was a significant linear correlation between the overall firing rate and overall E-box driven gene expression in *Per1*::GFP or *Per2*[−/−] SCN neurons and a loss of that correlation in *Per1*[−/−] SCN neurons, plotting these variables as 1 h averages over 24 h reveals a non-linear, cyclical trajectory. This indicates that the relationship between firing rate and the molecular clock over time is cyclical rather than linear.

Clock genes, including *Per1*, are widely expressed in the nervous system and therefore may be key regulators of neuronal activity in many brain circuits. Indeed, *Per1*[−/−] mice show alterations in many forms of neural plasticity including cocaine sensitization, LTP, and memory processing (*Abarca, Albrecht & Spanagel, 2002*; *Rawashdeh et al., 2014*). These results therefore expand upon the role of *Per1* in the generation of circadian rhythms and demonstrate that *Per1* in and of itself is key for the phasing of gene expression and neural activity in individual neurons.

### Funding

This work was supported by US National Institutes of Health grants R01 GM117650 (DGM), T32 MH064931 (Mark D. Wallace and DGM) and F31 NS082213 (JRJ). The funders had no role in study design, data collection and analysis, decision to publish, or preparation of the manuscript.

### Grant Disclosures

The following grant information was disclosed by the authors:
US National Institutes of Health: R01 GM117650, T32 MH064931, F31 NS082213.

### Competing Interests

The authors declare there are no competing interests.

### Author Contributions

- Jeff R. Jones conceived and designed the experiments, performed the experiments, analyzed the data, wrote the paper, prepared figures and/or tables, reviewed drafts of the paper.
- Douglas G. McMahon conceived and designed the experiments, wrote the paper, reviewed drafts of the paper.

### Animal Ethics

The following information was supplied relating to ethical approvals (i.e., approving body and any reference numbers):

All animal care and experimental procedures were conducted in concordance with Vanderbilt University's Institutional Animal Care and Use Committee guidelines.

### Data Availability

The raw data has been supplied as Data S1.

### Supplemental Information

Supplemental information for this article can be found online at http://dx.doi.org/10.7717/peerj.2297#supplemental-information.

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
