# Peer review of "The core clock gene Per1 phases molecular and electrical circadian rhythms in SCN neurons"

_PeerJ, doi:10.7717/peerj.2297_

## Round 0.1 · original submission · Minor Revisions

Three reviews from experts in the field were obtained for this manuscript. Two reviewers recommended minor revisions, mainly requiring clarifications and more detailed discussion of some key issues. The other reviewer recommended major revisions, raising some concerns about the experimental methodology. Based on these reviews and my own reading of this manuscript, I have recommended minor revisions. While all the comments in the three reviews need to be addressed, the authors should give the following points particular consideration:

1. One reviewer raised concerns about the significance of the current study being overstated. Please acknowledge that there is some existing knowledge about links between clock gene expression and neural activity.

2. All three reviewers raised concerns about the lack of discussion of in vivo experiments with Per1 knockout mice. These results should be discussed and rationalized with the current study.

3. One reviewer requests that the location of the SCN recordings should be clearly stated in the revision.

4. Reviewer 2 raises concerns about the limited expression of the dGFP construct. The authors should acknowledge and discuss this issue.

·

Basic reporting

The background is appropriate. The set-up is a little over-sold. I would argue that the relationship between the molecular clock and neural rhythm is not “unknown” but certainly not well understood. The figures are clear. The format appears to be appropriate with the raw data supplied.

Experimental design

The research question is well defined and addresses a gap in the literature. The experiments are well performed using methodology which is well established in the laboratory. There is no concern about the methods.

Validity of the findings

The observations are clear, novel, and will make a significant contribution to the field. The data supplied fits nicely with recent work (Kudo et al., 2015) and together these studies implicate the Period1 gene as playing a critical role in linking rhythmic gene expression and firing rate. The data is robust, significantly sound, and well controlled. The conclusion is well stated.

Additional comments

This is a very well written little gem of a study. While I know that novelty is not an issue with this journal, the findings are robust and impactful. Both the text and figures are clear.

I would recommend that the authors dial down the set-up. There is some body of work (including that of the authors) exploring the links between clock gene expression and neural activity. So the relationship is not "unknown" but definitely an area where more work is required.

I would ask that authors to acknowledge and briefly discuss how these results stand in contrast to the in vivo analysis of the Per1 mutants that show normal rhythms in wheel running behavior and SCN neural activity rhythms (Pendergast et al., 2009; Takasu et al., 2013).

I agree with the authors conclusion that this work suggest that Per1 plays a unique, non-redundant role in phasing gene expression and firing rate rhythms in SCN neurons.

Reviewer 2 ·

Basic reporting

This is a very clearly written manuscript that is easily follow and read. The background to the problem under investigation is succinctly accounted for in the Introduction. The main Results are presented in four good quality figures and these are all relevant to the findings.

Experimental design

The design of the studies is clearly described. The authors employ methods and approaches well-established in the PIs lab; brain slices were prepared from mice in which the activity of the Per1 promotor is reported by a destabilized green-fluorescent protein (dGFP) and the relationship between SCN neuronal activity (firing of action potentials as recorded in loose patch configuration) and dGFP expression examined. The authors also use mice with a variety of clock gene mutations to further assess the relationship between the molecular clock and SCN neuronal activity. Generally, there are sufficient details for the reader to understand how the experiments were conducted and there are no ethical concerns. An additional explanatory sentence on the Per1-/-;Per1::dGFP mice would be useful as it would be unclear to a reader unfamiliar with circadian biology as to how dGFP would be expressed if there is no Per1. One concern with the Experimental Design is that the authors do not describe where in the SCN the recordings were made (ie. Did they primarily target the ventral or central SCN?). This is important as the pattern of clock gene expression in the SCN is reported by several groups to vary in different subregions of the SCN.

Validity of the findings

In keeping with earlier research from the PI’s lab, the authors find that during the day, there is a positive correlation between the intensity in dGFP (as a proxy for Per1 expression) and the frequency of action potential firing of SCN neurons. Peak firing activity occurred around the middle of the day, while peak dGFP intensity occurred ~2h later. This positive relationship persisted in subjective daytime SCN recordings taken from animals that had been free-running in constant dark. This relationship was not disrupted by blockade of GABA signalling with gabazine, suggesting that neuronal state is cell autonomous and not dependent on intercellular signalling. In animals lacking Per1, the relationship was altered such that the peak in dGFP occurs some 7h after the peak in firing. In animals lacking Per2, the relationship was similar to that of genetically intact mice. Finally, in mice lacking both Per1 and Per2, the daytime relationship is lost. This series of investigation suggests that Per1 expression is pivotal to relationship between the molecular clock and SCN neuronal firing rate.

There is an impressive amount of work in this study—the authors are to be commended for this as these are not easy experiments. On the face of it, this seems a fairly straightforward outcome (for daytime recordings), however, there are some significant limitations that I think the authors need to acknowledge and expand on.

i) In the SCN, the dGFP construct is not as widely expressed as other reporters of the molecular clock, such as PER2::LUC. This suggests then that this construct under reports the numbers of ‘clock cells’ and that expression of dGFP has no bearing on the circadian pattern of firing for some neurons. This makes it difficult to state with absolute confidence that raised dGFP expression=increased firing rate (nb: dGFP-ve SCN neurons also increase their firing rate during the day and reduce it at night). Similarly, at night, dGFP expression is very low and this makes it difficult to distinguish dGFP+ve and –ve neurons. Also, at this time, SCN neurons can be silent (see work from Meijer lab) and not firing action potentials so the precise relationship is more complex than daytime relationship would indicate. The authors need to consider these limitations in the Discussion section.
ii) Mice lacking Per1 are behaviorally rhythmic (Cermakian et al., 2001) so it is unclear how the relationship between dGFP and SCN firing rate has been so overtly altered in these mice. This needs to be better explained.
iii) The relationship between neuronal activity and dGFP is also complicated for the early day when firing rate increases but dGFP does not. What happens to the relationship if the 24h cycle is analyzed in 4 or 6h segments?
iv) Have authors tried other markers of neuronal activity such as intracellular Ca2+? Inclusion of additional measures (assuming they has a similar trajectory) such as Ca2+ would strengthen their dataset.
v) Panels 1a, 2a, and 3a should be adjusted--there is no data post 24h so there is no cosine to fit.

Reviewer 3 ·

Basic reporting

1) The introduction is missing information about the behavioral phenotypes that have been previously reported in the Per1-/- and Per2-/- mice as well as the previously reported effects on circadian gene expression rhythms in these mice. This is important to understand the potential implications of the results presented, in the context of behavior and the previously published data.

Experimental design

1) The source of the Per1::GFP, Per1-/- and Per2-/- mice should be stated and the articles describing the generation of the mice should be cited.

Validity of the findings

This is an interesting paper that will contribute to our understanding of the relationship between the circadian rhythms in gene expression and the rhythms in electrical activity in the SCN.

1) On line 167 of the manuscript, the authors state "when the time to translate and fold GFP of approximately two hours is accounted for..." but do not provide any evidence or reference for this statement. A similar statement is repeated on line 227. The authors should provide data or a reference to support this statement.

2) On line 204, the authors state that there is "a statistical trend towards rhythmicity". How was this statistical trend determined?

3) The results presented in this manuscript demonstrate that knocking out Per1 or Per2 does not affect the molecular or the firing rhythmicity of the cells. This is consistent with previous reports and should be mentioned in the discussion.

4) Previous studies have demonstrated that Per1-/- and Per2-/- are both behaviorally arrhythmic. Interestingly, the results presented in this manuscript demonstrate that the phase relationship between the molecular and the electrical rhythms is only altered in Per1-/- mice. An interpretation of the results presented in this manuscript in the context of the previously published work and how it fits into the broader field of knowledge should be included.

5) On line 252, the authors discussed the models that have been proposed regarding the relationship between the molecular clock and the firing rates and how their new data fit into these models. An important conclusion that is missing in their discussion, though, is that in spite of the results from previous optogenetic experiments demonstrating that optogenetic manipulation can reset the molecular clockwork, the results presented in this manuscript demonstrate that firing rhythms are not sufficient to maintain the molecular clock.

---

## Round 0.2 · Minor Revisions

A second review was obtained from the Reviewer 2. Generally speaking, this reviewer was satisfied with the revision. In preparing their final version of the manuscript, I ask the authors to further clarify the two remaining issues raised by the reviewer:

1. Comparisons between fluorescence and firing rate.

2. References for SCN physiology.

Reviewer 2 ·

Basic reporting

The article is well-written are there are no concerns regarding the structure or of the figure result ion etc.

Experimental design

The experimental design details are adequate, although the comparisons in fluorescence and firing rate are still problematic (see below)

Validity of the findings

In general the findings are valid as reported, but there still remains a concern over the relationship between the Per1-driven fluorescence and spike activity. In Figure 1, it is clear that neurons with very low levels of fluorescence fire action potentials at ~6Hz, which is a similar frequency to those with a much higher level of fluorescence. Therefore, a linear fit seems at best forced. The authors should acknowledge the range of fluorescence at which firing frequencies >4 Hz (which is a typical day firing rate of a mouse SCN neuron) occur. This is an instance where a population average masks considerable individual variability. Indeed, the distributions from Figure 1b and 2b (Per1 null) do not appear to be that different, with exception of the absence of a small number of fast firing neurons in 2b. This needs to be suitably commented on.

Additional comments

The authors should address the concern raised above. Also, in the references, they cite Maywood et al. 2006 as showing that VIPergic signals are important for firing rate, but that study does not report on that aspect of SCN physiology. Other more suitable citations for that aspect of SCN physiology are Aton et al., 2005; Atkinson et al., 2011; Cutler et al., 2003.

---

## Round 0.3 · accepted · Accept

I am pleased to recommend this revised manuscript for publication.